# On Uncertainty Calibration and Selective Generation in Probabilistic Neural Summarization: A Benchmark Study

**Polina Zablotskaia**[*]    **Du Phan** [*]    **Joshua Maynez**    **Shashi Narayan**
**Jie Ren**    **Jeremiah Liu**
Google Research
{polinaz,phandu,joshuahm,shashinarayan,jjren,jereliu}@google.com

## Abstract

Modern deep models for summarization attains impressive benchmark performance, but they are prone to generating *miscalibrated* predictive uncertainty. This means that they assign high confidence to low-quality predictions, leading to compromised reliability and trustworthiness in real-world applications. Probabilistic deep learning methods are common solutions to the miscalibration problem. However, their relative effectiveness in complex autoregressive summarization tasks are not well-understood. In this work, we thoroughly investigate different state-of-the-art probabilistic methods' effectiveness in improving the uncertainty quality of the neural summarization models, across three large-scale benchmarks with varying difficulty using our newly introduced evaluation protocol. We show that the probabilistic methods consistently improve the model's generation and uncertainty quality, leading to improved selective generation performance (i.e., abstaining from low-quality summaries) in practice. We also reveal notable failure patterns of probabilistic methods widely-adopted in NLP community (e.g., Deep Ensemble and Monte Carlo Dropout), cautioning the importance of choosing appropriate method for the data setting.

## 1 Introduction

In recent years, autoregressive deep models for text summarization have achieved impressive performance. However, despite their success, these models often suffer from a critical flaw: they generate prediction with high confidence even when the quality of the summary is low (Xu et al., 2022). This can severely compromise the reliability and trustworthiness of the generated summaries in real-world applications. In the probabilistic forecast literature, such issue is known under the term *miscalibration*, i.e., the model's predictive confidence

is mis-aligned with its prediction quality. For example, in classification tasks, a model is said to be *miscalibrated* if for all test examples where it predicts with probability 0.9, the model's actual accuracy for these examples deviates far from 90% (Guo et al., 2017; Gneiting et al., 2007). Despite its practical importance, this notion of *uncertainty calibration* has received much less attention in the summarization literature until recently, with the proposed techniques mostly focusing on training deterministic models (Cao and Wang, 2021; Sun and Li, 2021; Zhao et al., 2022; Liu et al., 2022; Jung et al., 2021).

In the uncertainty literature, probabilistic deep learning has emerged as a principled approach to tackle model miscalibration while maintaining prediction quality (Nado et al., 2021). Intuitively, probabilistic DNNs generate multiple plausible predictions from its posterior predictive $\bar{p}_m(y|x) = \frac{1}{M} \sum_{m=1}^{M} p_m(y|x)$ and report the average, thereby mitigating the overconfidence of the individual model prediction. Although well-tested in classification tasks, the effectiveness of different state-of-art probabilistic methods in improving neural summarization models' uncertainty quality has been less explored. The existing study mostly focuses on a particular classic method (e.g., Monte Carlo Dropout or MCD) and tested on relatively simple datasets that doesn't fully capture the realistic usage (Gidiotis and Tsoumakas, 2022).

In this work, we address this by conducting a comprehensive investigation of the relative effectiveness of state-of-the-art probabilistic methods in improving the uncertainty quality of neural summarization models. We interrogate both classic approaches such as Monte Carlo Dropout (MCD) and Deep Ensemble (DE), and more recent state-of-art methods such as Batch Ensemble (BE) Spectral-normalized Gaussian process (SNGP) and their combinations that address the latency and quality caveats of the classic methods (Gal and

---

[*]Equal contribution.

Ghahramani, 2016; Lakshminarayanan et al., 2017; Liu et al., 2020; Wen et al., 2020). Furthermore, we evaluate method performance across multiple benchmarks of varying difficulty to ensure the practical relevance of our result, and to uncover potential failure patterns of different approaches. Our contributions are:

• We adapt the various probabilistic deep learning methods to the Pre-trained Language Model (PLM) setup and conduct an extensive study on their effect on both uncertainty and prediction aspects of model performance.

• We propose a new evaluation protocol to measure the uncertainty calibration performance of summarization models, tailored toward domain-specific quality scores (e.g., ROUGE).

• We show that using probabilistic methods generally leads to improved summarization and calibration performance, and consequently improved selective generation. We also discuss the failure patterns of the popular methods such Deep Ensembles (Lakshminarayanan et al., 2017) and Monte Carlo Dropout (Gal and Ghahramani, 2016).

## 2   Related work

**Probabilitistic Learning for Seq2seq Models.** Developed primarily in the context of discriminative models, the state-of-art probabilistic approaches can be applied to large neural models without sacrificing performance (Gal and Ghahramani, 2016; Lakshminarayanan et al., 2017; Wen et al., 2020; Liu et al., 2020). Recently, however, initial investigations into unsupervised uncertainty estimation for structured prediction have appeared, primarily focusing on more basic approaches such as Monte Carlo Dropout (MCD) or Deep Ensemble (DE) (Xiao and Wang, 2019; Wang et al., 2019; Fomicheva et al., 2020; Malinin and Gales, 2021; Lin et al., 2022), with a few work looking into summarization tasks (Xu et al., 2020; Gidiotis and Tsoumakas, 2022). In comparison, this work focuses on an unbiased evaluation of a wide range of state-of-the-art probabilistic methods on tasks with varying difficulty, and reveals failure patterns of classic approaches such as MCD and DE.

**Calibration Technique in Language Processing.** Guo et al. (2017) proposed improving calibration of document classifier using of temperature scaling. Müller et al. (2019) and Wang et al. (2020a) explored improving calibration in neural machine translation using label smoothing. Desai and Dur-

rett (2020) noted that calibration methods can be used to improve the accuracy of pre-trained language models. Jung et al. (2021) proposed a novel training approach to improve calibration by minimizing a combined loss of cross-entropy and calibration. In the summarization literature, (Cao and Wang, 2021; Xu et al., 2022; Sun and Li, 2021; Zhao et al., 2022; Liu et al., 2022) explored calibrating model probability using contrastive learning approaches. Most of these techniques focus on deterministic models. They are orthogonal to and can be combined with the probabilistic approaches we explore in this work.

## 3   Methods

Probabilistic methods have been adopted to increase the reliability of pre-trained language models. Plex paper (Tran et al., 2022) provided a nice survey on the robustness of uncertainty methods on text classification tasks. In this study, we opted for the methods that are widely recognized and used. Section A.7 contains in-depth details, here we provide a general overview:

**Single-model Methods:**
• **Deterministic Baseline** - we use the base T5 model [1] (Raffel et al., 2020) as the baseline model.
• **Monte Carlo Dropout (MCD)** (Gal and Ghahramani, 2016) which estimates uncertainty using the Monte Carlo average of 10 dropout samples. Those samples are generated using the same model and parameters but with different random seeds at dropout.
• **Batch Ensemble (BE)** (Wen et al., 2020) - an ensemble method which has much lower computational costs comparing to MC Dropout and Deep Ensemble. We replace the last transformer's MLP block by a batch ensemble block with ensemble size be 5.[2]
• **Spectral-normalized Neural Gaussian Process (SNGP)** (Liu et al., 2020) - a recent state-of-the-art approach which improves uncertainty quality by transforming a neural network into an approximate Gaussian process model. The Gaussian Process last layer is able to reflect the distance between a test example and the training set, hence potentially be helpful in improving calibration.
• **SNGP+MCD** which is the MC Dropout on top of an SNGP model;

---
[1] All methods can be applied to larger models.
[2] BE requires more memory on a single machine, so we keep the ensemble size below 10.

**Multi-model Methods:**

• **Deep Ensemble (DE)** (Lakshminarayanan et al., 2017) which trains 10 deterministic models individually and averages all. We use the same model architecture but changing the initial seeds.

• **Gaussian Process Ensemble (SNGP+DE)** is the combination of deep ensemble and SNGP.

For all methods, we use the official base T5 checkpoint, which are pretrained on a large corpus like C4 (Raffel et al., 2020). We then fine-tune the parameters on summarization tasks. To generate prediction from the model posterior in all, we perform beam inference with respect to the model's conditional posterior mean probability, i.e., $\bar{p}(y_t|y_{<t}, x) = \frac{1}{M}\sum_{m=1}^{M} p_m(y_t|y_{<t}, x)$, where $M = 10$ is the number of samples from model posterior (for the deterministic baseline and **SNGP**-only method $M = 1$). To quantify model uncertainty, we consider the length-normalized predicted log-probabilities following previous work, i.e., $u(y|x) := \frac{1}{T}\sum_{t=1}^{T} \bar{p}(y_t|y_{<t}, x)$ (Wu et al., 2016; Liu et al., 2022) where $x$ is the input sequence, $y$ is the output sequence, $y_t$ is the $t$-th token of $y$, and $T$ is the length of the sequence, i.e. the number of tokens in $y$.

## 4 Experiments

For the first time, we benchmark the probabilistic uncertainty calibration methods. We use our proposed evaluation protocol, consisting of assessing the ROUGE quality improvements, measuring uncertainty calibration, which includes our newly proposed *sequence-level* Expected Calibration Error (ECE), assessing rank correlation and finally analysing selective generation via abstention as a measure of language model calibration. We evaluate all methods on three datasets: **XSUM** (Narayan et al., 2018), **CNN/DailyMail** (Hermann et al., 2015; See et al., 2017) and **RedditTIFU-long** (Kim et al., 2019) due to their diversity in abstractiveness, lengths, domain and style (see Section A.6 for additional details). For all experiments we use beam search decoding, which was adapted to work with ensemble generation. We have also adapted **SNGP** and **BE** algorithms to work with the sequence generation and the corresponding loss (see Section A.7 for more details).

### 4.1 ROUGE with Probabilistic Methods

We first study the effectiveness of different probabilistic methods on summary prediction by compar-

ing them with the baseline deterministic model. We use ROUGE-1/2/L (Lin, 2004) to measure general summarization quality. As shown in Table 1, we observe the consistent improvement of the ROUGE scores in probabilistic models compared to baselines. For single model methods, SNGP achieves the highest average ROUGE scores over the three datasets. Other probabilistic methods also show promising performance: SNGP+MCD is ranked the second top regarding ROUGE-1, and BE is ranked the second top regarding ROUGE-2 and the top regarding ROUGE-L. For multiple model methods, SNGP+DE improves over the deterministic DE. Comparing multiple model methods with single model methods, DE and SNGP+DE generally have higher ROUGE scores than single model methods.

| ROUGE-1↑ | | | | | |
|---|---|---|---|---|---|
| Method | XSUM | CNN/DM | Reddit | Average↑ | Average Rank↓ |
| Base | 40.83 | 41.19 | 26.14 | 36.05 | 2.67 |
| Base nucleus | 40.03 | 40.66 | 24.54 | 35.08 | 5.33 |
| SNGP | 40.79 | 41.76 | 26.08 | **36.21** | **2.33** |
| MCD | 40.31 | 40.68 | 24.27 | 35.09 | 5.00 |
| SNGP+MCD | 40.90 | 41.49 | 24.60 | 35.66 | 2.33 |
| BE | 41.21 | 41.22 | 23.36 | 35.26 | 3.33 |
| DE | 41.51 | 41.20 | 26.65 | 36.45 | 1.67 |
| SNGP+DE | 42.14 | 41.99 | 26.57 | **36.90** | **1.33** |
| ROUGE-2↑ | | | | | |
| Method | XSUM | CNN/DM | Reddit | Average↑ | Average Rank↓ |
| Base | 19.14 | 19.77 | 7.76 | 15.56 | 2.67 |
| Base nucleus | 18.15 | 18.89 | 6.47 | 14.50 | 6.00 |
| SNGP | 18.91 | 20.19 | 7.76 | **15.62** | **2.00** |
| MCD | 18.63 | 19.78 | 7.12 | 15.18 | 4.33 |
| SNGP+MCD | 18.91 | 20.33 | 7.00 | 15.41 | 3.00 |
| BE | 19.41 | 19.81 | 7.31 | 15.51 | 2.33 |
| DE | 19.84 | 19.77 | 8.21 | **19.81** | 1.67 |
| SNGP+DE | 20.35 | 20.49 | 7.77 | 16.20 | **1.33** |
| ROUGE-L↑ | | | | | |
| Method | XSUM | CNN/DM | Reddit | Average↑ | Average Rank↓ |
| Base | 33.76 | 38.54 | 21.31 | 31.20 | 2.33 |
| Base nucleus | 33.04 | 38.13 | 19.79 | 30.32 | 5.67 |
| SNGP | 33.53 | 39.12 | 21.17 | **31.27** | 2.67 |
| MCD | 33.21 | 38.09 | 20.06 | 30.45 | 5.00 |
| SNGP+MCD | 33.59 | 38.97 | 20.00 | 30.85 | 3.00 |
| BE | 34.06 | 38.50 | 20.81 | 31.12 | **2.00** |
| DE | 34.38 | 38.54 | 21.72 | **31.55** | **1.33** |
| SNGP+DE | 34.92 | 36.55 | 21.18 | 30.88 | 1.67 |

Table 1: ROUGE scores and ranking of different probabilistic methods across all datasets. Probablistic methods consistently outperform base model, and SNGP-family models generally lead to strong performance. For this experiment we additionally conduct evaluation of the Base model with nucleus sampling ($p = 0.5$), which sometimes improves model performance by truncating the less reliable tail of the distribution, however it doesn't change the model calibration. The best and second-best results are denoted by bold and underlined formats, respectively.

### 4.2 Measuring Uncertainty Calibration in Summarization

We now study model's uncertainty calibration quality. We consider both the classic metric Expected

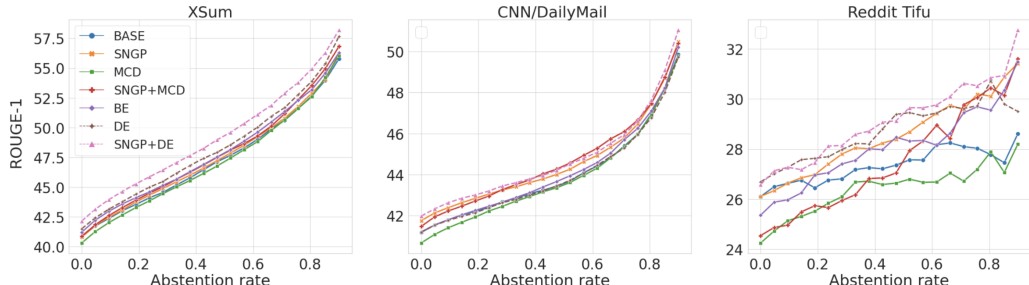

Figure 1: ROUGE vs Abstention Curve for different probabilistic methods. **Abstention rate** $\alpha$ denotes the percentage of examples that were excluded, after ranking according to log-probabilities. For single model methods (solid lines), SNGP+MCD models have generally higher ROUGE scores in CNN/DM, and in regions of $\alpha > 0.6$ in XSUM and Reddit. For multi-model methods, SNGP+DE generally outperforms DE in all the three datasets. See Figure 4 for the results on other ROUGE scores.

Calibration Error (ECE), and also the uncertainty score's Spearman's rank correlation with domain-specific quality scores tailored for summarization (e.g., ROUGE).

**ECE.** In order to evaluate whether the model estimated probabilities have been more calibrated we assess the difference in expectation between confidence and accuracy using ECE metric (Naeini et al., 2015):

$$ECE = \sum_{k=1}^{K} \frac{|B_k|}{n} |\text{conf}(B_k) - \text{acc}(B_k)|,$$

where we split the interval $(0, 1]$ into $K$ equal-size bins and define $B_k$ to be the set containing the indices of examples which have predicted probability lie in the $k$-th bin: $B_k = \left\{ i | \hat{p}_i \in \left( \frac{k-1}{K}, \frac{k}{K} \right] \right\}$, where the average accuracy and confidence within each bin are defined as $\text{acc}(B_k) = \frac{1}{|B_k|} \sum_{i \in B_k} I(\hat{y}_i = y_i)$ and $\text{conf}(B_k) = \frac{1}{|B_k|} \sum_{i \in B_k} \hat{p}_i$.

In auto-regressive prediction, $\hat{y}_i$ can be a sequence or a token,[3] which corresponds to two different metrics *sequence-level ECE* and *token-level ECE* respectively. When computing the sequence-level ECE, we cast the problem into a binary classification task, where the probability $\hat{p}$ of a predicted sequence is the production of probabilities of all tokens of that sequence. Regarding the token-level ECE, other work (e.g. Wang et al. (2020b)) uses translation edit rate (Snover et al., 2006) to relax the condition that the tokens under consideration need to be at the same position. In our work, we say that a predicted token is correct if it matches the target token at the same position in the target sequence.

---

[3] During evaluation, we compute token probabilities from the highest scoring beam sequence.

As shown in Table 2, across all methods, SNGP+MCD and SNGP+DE generally leads to lower ECE in single model and multi-model methods respectively, suggesting SNGP helps to reduce ECE.

| Method | Sequence-level ECE $\times 10^{-3}$ ($\downarrow$) | | | | Token-level ECE $\times 10^{-1}$ ($\downarrow$) | | | |
| --- | --- | --- | --- | --- | --- | --- | --- | --- |
| | XSUM | CNN/DM | Reddit | Average | XSUM | CNN/DM | Reddit | Average |
| Base | 2.70 | 0.28 | 0.32 | 1.10 | 5.89 | 7.69 | 4.56 | 6.05 |
| SNGP | 3.47 | 0.52 | 1.13 | 1.71 | 5.97 | 7.71 | 5.26 | 6.31 |
| MCD | 1.02 | 0.18 | 0.05 | 0.42 | 4.54 | 6.68 | 3.05 | **4.76** |
| SNGP+MCD | 0.93 | 0.11 | 0.13 | **0.39** | 4.54 | 6.69 | 3.44 | 4.89 |
| BE | 2.65 | 0.64 | 0.44 | 1.24 | 5.95 | 8.04 | 5.01 | 6.33 |
| DE | 1.89 | 0.13 | 0.63 | 0.88 | 5.58 | 7.78 | 4.82 | 6.06 |
| SNGP+DE | 0.83 | 0.36 | 0.90 | **0.70** | 5.39 | 7.62 | 5.15 | **6.05** |

Table 2: ECE on sequence and token levels of different probabilistic methods across all datasets. SNGP+MCD and SNGP+DE generally leads to lower ECE in single model and multi-model methods, respectively.

**Rank Correlation with Quality Score.** We investigate how the Spearman's rank correlation between the log-probabilities and ROUGE changes with calibration. Overall we see a general trend demonstrating the calibration increases the correlation, as shown in Figure 2. For the ROC-AUC scores please refer to the section A.1.

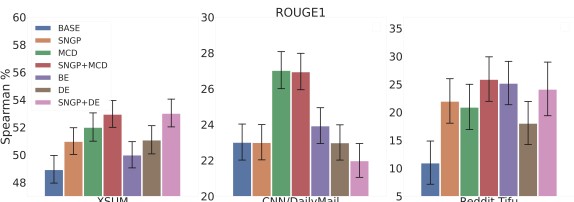

Figure 2: Spearman's rank correlation between the length-normalized log-probabilities and the ROUGE-1. We compute the error bars using bootstrap standard deviation technique. The best and second-best results are denoted by bold and underlined formats, respectively.

### 4.3 Selective Generation via Abstention

Selective generation refers to the procedure to selectively generate higher-quality outputs while abstain the low-quality outputs (Ren et al., 2022). It evaluates the models' uncertainty calibration, since

a well calibrated model is expected to have high uncertainty for low-quality outputs such it can be used to select examples to abstain. We use the *ROUGE vs Abstention Curve* [4] to compare methods: specifically, at a given abstention rate $\alpha$, we remove the lowest $\alpha$-fraction uncertain examples and compute the average quality of the remaining examples, as shown in Figure 1. For single model methods (solid lines), SNGP+MCD models have generally higher ROUGE scores in CNN/DM, and in regions of $\alpha > 0.6$ in XSUM and Reddit. For multi-model methods, SNGP+DE generally outperforms DE in all the datasets.

**Failure Patterns.** When comparing multi-model methods with single model methods, we observe that XSUM and Reddit both have multi-model methods outperforming single model methods, but CNN/DM does not benefit from using multi-model methods. This difference can be explained by the fact that CNN/DM is an simpler task that is more extractive in nature, and a single model already performs well and relatively calibrated. In this case, using a deep ensemble can in fact lead to under-confidence (Rahaman et al., 2021). Furthermore, in Reddit dataset, MCD-family methods seem to lead to severe degradation of summarization quality. Note that Reddit is a more challenging task with much greater linguistic diversity when compared to XSUM and CNN/DailyMail, cautioning the use of MCD method in challenging test environments where a single model does not perform well.

## 5  Conclusion

We have adapted the most popular probabilistic calibration methods to the PLMs use-case and have conducted a novel and extensive study of the calibration effect on PLMs uncertainty and summarization quality. We proposed a novel evaluation protocol for the uncertainty estimation in PLMs. We demonstrated that probabilistic calibration methods can have a positive impact on the quality of generated summaries and increase reliability of the models. Our main findings can be summarized in two points:

- When there is no time and memory constraints, our results point out that it is best to choose Deep Ensembles in combination

with SNGP as an approach to language model calibration, since they are effective in terms of reducing the ECE and improving the quality of summarization.

- Even when calibration methods appear to be effective in reducing ECE it may not necessarily suggest that they will be effective in improving ROUGE or other language generation quality metrics.

We hope to see our efforts making a significant contribution to the improvements of the reliability of PLMs, enabling future researchers to effectively leverage probabilistic calibration methods in the development and analysis of these models.

## 6  Limitations

In our paper we investigated the effect of most common and widely used probabilistic deep learning methods on Pre-trained Language Models. We observed a positive effect of calibration on the variety of metrics, however it is important to address the limitations of this work. Even though we observed the improvements on multiple metrics, we believe more work needs to be done to fully understanding the interplay between classic probabilistic deep learning methods that were traditionally applied in the context of classification and the unique set up the language models pose: autoregressive generation and mismatch between the learning and inference. Above mentioned unique properties of PLMs make it harder to align the model predictive probability with the distribution of the data. In our work we have done very basic adaptation of the existing methods to the PLM setting, but in future more work needs to be done to address these differences.

## 7  Ethical impact

Our work directly contributes to the topic of reliable deep learning. We believe our work should positively affect the scientific community, since we address one of the main problems that often occurs in the machine learning research: how to make the models more reliable and trustworthy. We hope that in long run we can arrive at a standardized benchmark set of techniques that can help the NLP community develop PLMs that are universally trustworthy.

---

[4] Different from the performance vs data retention curves in Filos et al. (2019), we employ log probability rather than predictive entropy as the metric for data retention.

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

# A Appendix

## A.1 ROC-AUC scores

| Metric | Method Type | Method | XSUM | CNN/DailyMail | Reddit Tifu |
|---|---|---|---|---|---|
| ROUGE1 | Single-model | Base | 73.16 | 60.38 | 60.28 |
| | | SNGP | 74.34 | 59.96 | 65.01 |
| | | MCD | 75.06 | 62.29 | 69.98 |
| | | SNGP+MCD | 75.14 | 62.05 | 67.60 |
| | | BE | 73.30 | 60.45 | 64.08 |
| | Multi-model (10) | DE | 73.90 | 59.97 | 64.07 |
| | | SNGP+DE | 75.19 | 59.25 | 63.65 |
| ROUGE-2 | Single-model | Base | 73.00 | 60.22 | 59.93 |
| | | SNGP | 73.46 | 60.47 | 63.96 |
| | | MCD | 74.57 | 62.38 | 61.27 |
| | | SNGP+MCD | 74.58 | 63.22 | 63.58 |
| | | BE | 72.82 | 59.90 | 63.23 |
| | Multi-model (10) | DE | 74.03 | 59.93 | 62.59 |
| | | SNGP+DE | 74.52 | 60.27 | 61.79 |
| ROUGE-L | Single-model | Base | 71.75 | 59.25 | 61.34 |
| | | SNGP | 72.86 | 59.06 | 59.44 |
| | | MCD | 73.86 | 61.38 | 66.33 |
| | | SNGP+MCD | 74.61 | 61.35 | 62.99 |
| | | BE | 72.45 | 60.03 | 64.42 |
| | Multi-model (10) | DE | 72.85 | 58.78 | 65.05 |
| | | SNGP+DE | 74.15 | 59.51 | 61.39 |

Table 3: We measure the Area Under Curve values, when using the log-probabilities as a signal for "good/bad" summaries. Good and bad summaries are defined by a threshold $\theta$ we impose on the metric, i.e. when metric is above certain $\theta$ then we treat the summary as good and when it is below we treat it as a bad summary. We used the following $\theta$ for ROUGE1, ROUGE-2 and ROUGE-L correspondingly: 40, 15, and 30[5].

## A.2 Spearman's rank correlation ROUGE-2 and ROUGE-L

Spearman's rank correlation for the rest of the metrics can be found on Figure 3.

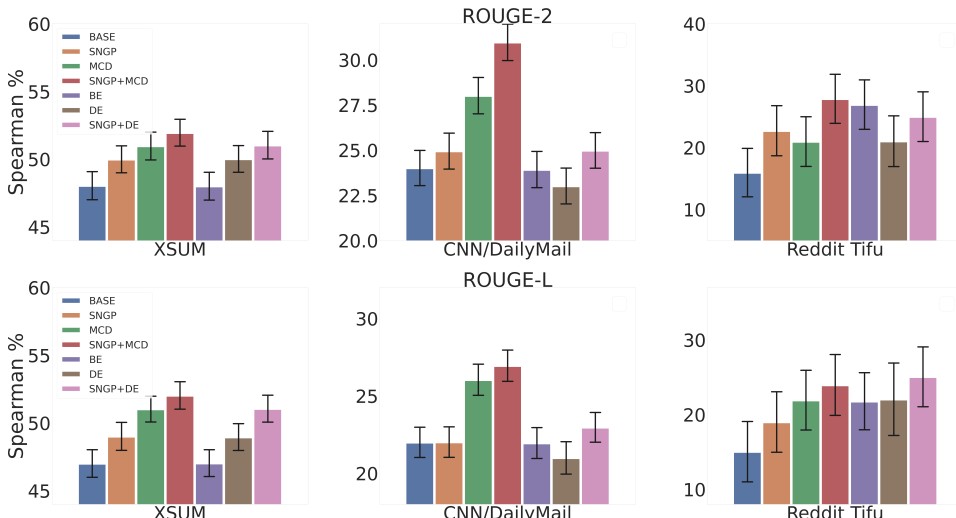

Figure 3: Spearman's rank correlation between the length-normalized log-probabilities and the ROUGE-2 and ROUGE-L.

## A.3 Abstention plots

We demonstrate the abstention plots for the rest of the metrics on Figure 4.

## A.4 Experimental details

We ran all the experiments on the T5 base model (220 million parameters) using open-sourced T5X framework (Roberts et al., 2022). We used the TPU v3 chips (Jouppi et al., 2020) for all experiments. Reported metric results are collected from a single evaluation run, unless error bars are provided or

---

[5]The values were selected as round numbers on the SoTA performance circa 2020, when NLG fluency was human-like: https://paperswithcode.com/sota/text-summarization-on-x-sum.

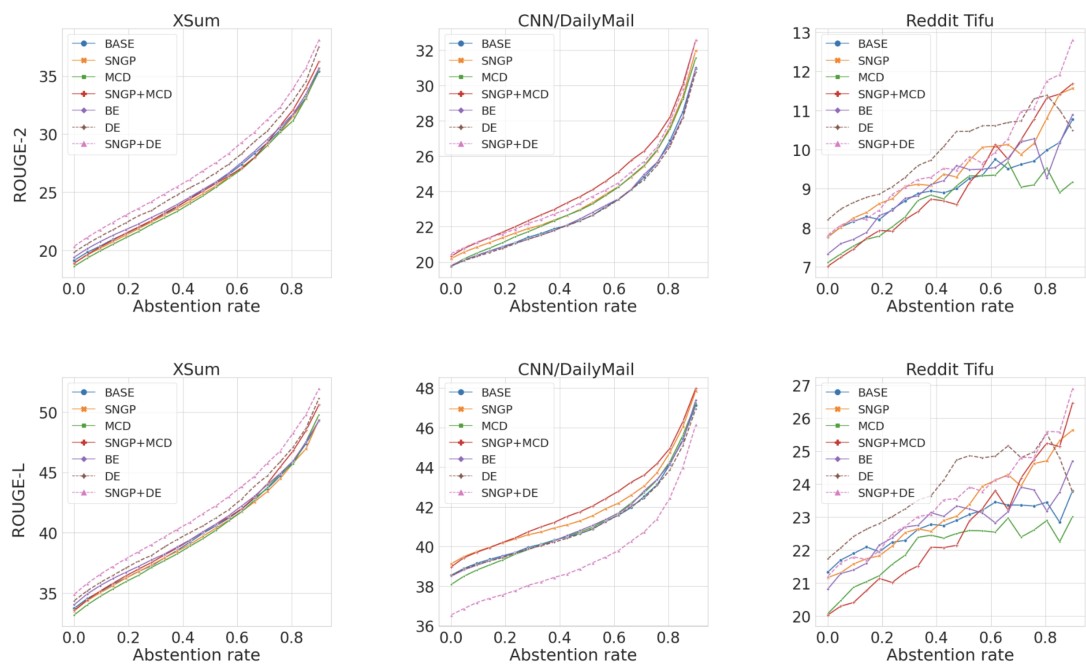

Figure 4: ROUGE-2 and ROUGE-L abstention plots.

stated otherwise. To select each model checkpoint we ran a hyperparameter sweep to find the best set of parameters. Parameters we sweeped over were: checkpoint step, leaning rate, SNGP mean field factor, number of checkpoints for the ensembles and number of training steps.

In all the experiments we use beam search (except where we stated "nucleus sampling") as a decoding method and we use beam_size= 3. For the MCD we used dropout_rate = 0.1 everywhere. Covariance matrix momentum in SNGP was set to 0.999. For the XSum the best mean field factor $10^{-4}$ and for CNN and Reddit it was $10^{-6}$.

## A.5 Qualitative results

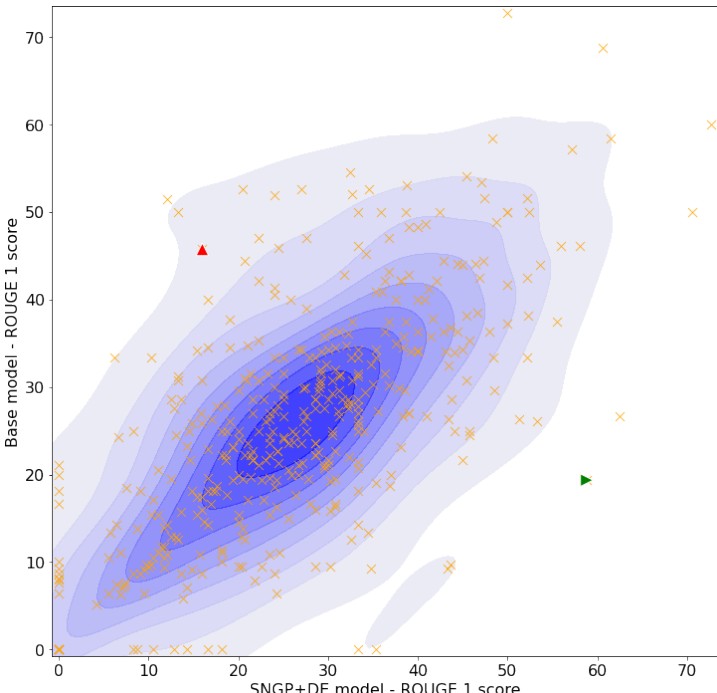

Figure 5: Scatter plot for ROUGE 1 scores of SNGP+DE and Base models on RedditTifu task. Detailed contents of ▶ and ▲ symbols can be founded in Tables 4 and 5 respectively.

## A.6 Dataset

**XSUM** (Narayan et al., 2018) consists of 227k BBC articles from 2010 to 2017 with a single sentence highly abstractive summary. The average length of each input document is 687 words, whereas the summary is 53 words long. Sometimes the summary contains information not present in the article (Maynez et al., 2020).

**CNN/DailyMail** (Hermann et al., 2015; See et al., 2017) contains 313k articles from the CNN and Daily Mail newspapers with bullet point summaries. The summaries are on average 3-4 sentences and relatively extractive. The average length of each input document is 375 words, whereas the summary is 21 words long.

**RedditTIFU-long** (Kim et al., 2019) contains 42k posts of informal stories from sub-reddit TIFU from 2013-Jan to 2018-Mar with author written summaries. The style and length of the summaries are very diverse. The average length of each input document is 433 words, whereas the summary is 23 words long.

## A.7 Methods details

In this section, we delve into the methods, offering a comprehensive explanation of their individual characteristics and how their combined properties can best help in model calibration. Deep Ensembles, Monte Carlo dropout, SNGP, and Batch Ensemble are all techniques used in the field of machine learning to improve the performance and robustness of neural network models. While they share the goal of calibrating the model predictions, they differ in their underlying principles and methodologies. Let's discuss each method and highlight their fundamental differences.

The main idea behind Monte Carlo Dropout, Deep Ensembles and Batch Ensembles is to get an approximation of the predictive posterior of a model via computing a mean of token probabilities from several unique models. Once we have access $M$ predictions we can compute the predictive posterior the

| |
|---|
| **Input:** summarize: my family was out on our porch, having some drinks and horderves before giving dad his father's day gift. at some point my mom notices one of our neighbors walking down the street. she points out that hes wearing these funny shoes and i turn around to see him shuffling his feet while wearing these giant slippers that look like part of a goofy costume. as he's crossing the street to check on his daughter playing at her friends house my dad yells "love those shoes!" and the guy waves and responds "thanks, they're from her," while pointing to his daughter. him and his daughter exchange a few words and he turns around to walk back home. you have to imagine this guy moving his body side to side while walking, dragging his one foot and swinging the other around, making it look like the slippers were heavy (since they were quite large). as he passes in front of our house again, i decide to poke a little fun at the way he walked in the slippers and yelled with a chuckle, "hey y'know you walk a little funny in those!" to which he replied, "heh, i dont have much of a choice," and continues walking with his head down. in my brain, he was saying "there's no other way to walk in these damn things," so i laughed audibly so he could hear i appreciated the response and attitude about the silly present. once he's inside, my mom turns to me and says, "yknow he used to weigh like 300 pounds, he's lost over 120. he also has cerebral palsy so he's always kinda walked with a limp." |
| **Target:** thought a guy's shoes were the reason he walked funny, turns out he has cerebral palsy. |
| **Base model:** i made fun of a guy walking in giant slippers on father's day. |
| **SNGP+DE model:** i laughed at the way a guy walked in his slippers, turns out he has cerebral palsy. |

Table 4: An example where SNGP+DE model gives better ROUGE1 score than Base model. This is annotated by ▶ symbol in Figure 5.

following way:

$$\bar{p}(\boldsymbol{y}|\boldsymbol{x},\mathcal{D}) = \mathbb{E}_{q(\theta)}[p(\boldsymbol{y}|\boldsymbol{x},\boldsymbol{\theta})] \approx \frac{1}{M}\sum_{m=1}^{M} p_m(\boldsymbol{y}|\boldsymbol{x},\boldsymbol{\theta}^{(m)}), \boldsymbol{\theta}^{(m)} \sim q(\boldsymbol{\theta}) \approx p(\boldsymbol{\theta}|\mathcal{D}) \quad (1)$$

Here, $\theta$ are the model parameters, $\boldsymbol{y}$ is the model prediction and $\boldsymbol{x}$ is the model input (for simplicity we consider $\boldsymbol{y}$ to be a whole sentence instead of an individual token, that are dependent on the previous tokens). $q(\boldsymbol{\theta})$ is the parameters prior distribution, in our case we used standard normal distribution to initialize the model parameters. Once we have an approximation of the model's predictive posterior we can estimate the expected uncertainty:

$$u(\bar{p}(\boldsymbol{y}|\boldsymbol{x},\mathcal{D})) = \mathcal{H}[\bar{p}(\boldsymbol{y}|\boldsymbol{x},\mathcal{D})] = \mathbb{E}_{\bar{p}(\boldsymbol{y}|\boldsymbol{x},\mathcal{D})}[-\ln \bar{p}(\boldsymbol{y}|\boldsymbol{x},\mathcal{D})] = -\sum_{\boldsymbol{y}\in\mathcal{Y}} \bar{p}(\boldsymbol{y}|\boldsymbol{x},\mathcal{D})\ln \bar{p}(\boldsymbol{y}|\boldsymbol{x},\mathcal{D}) \quad (2)$$

The uncertainty is essentially the Shannon entropy, and therefore can be easily calculated, once we have the values and probabilities across the whole space of possible outputs $\mathcal{Y}$, in practice we compute everything at the token level, and $\mathcal{Y}$ just becomes the models' vocabulary.

• **Monte Carlo Dropout (MCD)** (Gal and Ghahramani, 2016) is a technique that leverages dropout regularization during training inference. Dropout is a regularization technique that randomly sets a fraction of neural network units to zero during training. During inference, MCD involves performing multiple forward passes with dropout enabled (but with different random seeds) and averaging the predictions (more precisely, the probabilities of all tokens in the vocabulary). By sampling multiple predictions, MCD provides a measure of model uncertainty or confidence in its predictions. In our case MCD estimates uncertainty using the Monte Carlo average of 10 dropout samples.

• **Batch Ensemble (BE)** (Wen et al., 2020) - an ensemble method which has much lower computational costs comparing to MC Dropout and Deep Ensemble. Batch Ensemble is a technique that involves training multiple neural network models simultaneously within a single batch. Each model in the ensemble receives a different subset of the data batch, and the models share their weights at the end of each training iteration. By training models on different data subsets, Batch Ensemble encourages diversity and reduces overfitting. In the experiments, we replaced the last transformer's MLP block and the last dense layer by batch ensemble blocks with ensemble size be 5.

| |
|---|
| **Input:** summarize: i[f] have small hairs on my lip and waxing doesn't seem to work because of how tiny and thin they are. shaving doesn't help very much since it causes farther irritation. when i was feeling extra self concious after trimming the hairs on valentines day, my fiancé brought up trying the cream hair removal. |
| i went on amazon and bought the veet hair removal cream. last night, i got it in the mail and read the precautions. i saw not to use it on the face but like an idiot, i thought to ignore it (as does every story about the hair removal cream). it totally did the trick and my lip is hairless. i felt a bit of burning and irritation on the lip after but it went away after using a bit of bio oil. didn't think about it all night. |
| this morning on the other hand i woke up with more pain that you would feel after a burn. there was a small patch of skin breakdown and irritation to the left of my lip and a bit of redness on my upper lip but nothing more. i covered it up with makeup and it seemed to have done the trick. |
| fast forward about 8hrs. i now have small pin sized scabs all across my upper lip and pain. i look like a 15 year old boy who doesn't know how to shave or someone with uncontrollable herpes cold sores. ontop of that i got a venus razors ad while i write this on my smart phone to rub it in some more that i should've used a razor. the next week will be ugly. |
| **Target**: used hair remover cream on face, now have a chemical burn on the upper lip |
| **Base model:** i used veet hair removal cream on my lip and now i have small scabs all over my upper lip. |
| **SNGP+DE model:** veet hair removal cream made me look like an idiot. |

Table 5: An example where SNGP+DE model gives worse ROUGE1 score than Base model. This is annotated by ▲ symbol in Figure 5.

• **Deep Ensemble (DE)** (Lakshminarayanan et al., 2017) which trains 10 deterministic models individually and averages all. Deep Ensembles involve training multiple neural network models independently and then combining their predictions to make a final prediction. Each model in the ensemble is typically trained with different initialization, architectures, or subsets of the training data. The main idea is that the diverse models capture different aspects of the data and, when combined, produce more accurate and robust predictions. Deep Ensembles are computationally expensive since they require training and storing multiple models.

The above mentioned methods are very good at increasing the representation diversity, i.e. learning the multiple modes of the underlying data distribution. However, when dealing with uncertainty it is also important that model has a distance awareness, i.e. the property of the model that allows it to quantify the distance of $x_{test}$ and $X_{train}$ in the input space $||x_{test} - X_{train}||_x$. Typical Neural Networks are not distance aware. Spectral-normalized Neural Gaussian Process was proposed as a solution for this problem:

• **SNGP** (Liu et al., 2020) - a recent state-of-the-art approach which improves uncertainty quality by transforming a neural network into an approximate Gaussian process model. The Gaussian Process last layer is able to reflect the distance between a test example and the training set, hence potentially be helpful in improving calibration. The model architecture includes a neural network that maps inputs to a lower-dimensional feature space, and a Gaussian process layer that models the target variable using the extracted features. We adapted SNGP to the sequence generation by letting all pre-logits (i.e. inputs of the last dense layer) in a sequence go through the same GP layer.

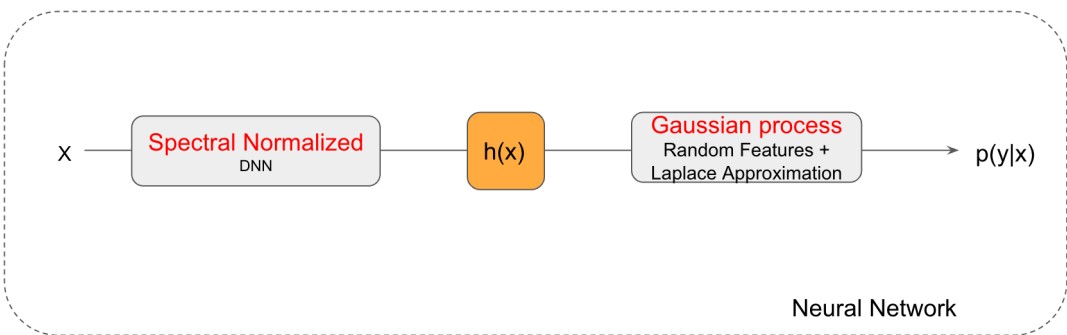

Figure 6: Architecture changes for the SNGP model. Spectral normalization enforces bi-Lipschitz smoothness, which discourages inputs that are far away in input space get mapped close in hidden space. For GP layer confidence is a function of distance from the training data.

In summary, the fundamental differences among these techniques lie in their approaches to ensemble learning, uncertainty estimation, and distance awareness. Deep Ensembles train multiple models independently and combine their predictions, Monte Carlo dropout incorporates dropout during training and testing, Batch Ensemble trains multiple models within a single batch and shares their weights and SNGP combines neural networks with Gaussian processes. Each method offers unique advantages and can be employed based on the specific requirements of the problem at hand, our experiments showed that the best results can be achieved by combining the SNGP with Deep Ensembles, which gives us the best of both worlds, i.e. representation diversity and distance awareness. See Table 6 for the summaries of different methods properties.

| Method | Distance awareness | Number of models | Number of inference runs | Simplicity of implementation | Representation diversity |
|---|---|---|---|---|---|
| MCD | ✗ | 1 | 10 | ✓ | ✓ |
| BE | ✗ | 1 | 1 | ✗ | ✓ |
| SNGP | ✓ | 1 | 1 | ✗ | ✗ |
| SNGP+MCD | ✓ | 1 | 10 | ✗ | ✓ |
| DE | ✗ | 10 | 10 | ✓ | ✓✓ |
| SNGP+DE | ✓ | 10 | 10 | ✗ | ✓✓ |

Table 6: Comparing studied calibration techniques against selected properties.

Finally, in order to compare the algorithms it is helpful to look at the total complexity:

- SNGP, BE: time complexity $O(T)$, space complexity $O(S)$,

- MCD, SNGP+MCD: time complexity $O(M \times T)$, space complexity $O(S)$,

- DE, SNGP+DE: time complexity $O(M \times T)$, space complexity $O(M \times S)$,

where $T$ and $S$ are time and space complexity of the base model and $M$ is the ensemble size. To reduce the memory required in DE models, we compute the predictions (i.e. the probabilities of all tokens in the vocabulary) of each model sequentially before averaging them. For that reason, the DE models have a time complexity $O(M \times T)$. The actual time and memory required for each method depend on the TPU topology, and they roughly align with the time and space complexity mentioned above.