# OpenReview forum: "On Uncertainty Calibration and Selective Generation in Probabilistic Neural Summarization: A Benchmark Study"
_EMNLP/2023/Conference — EMNLP 2023 Findings_

### Official Review · Reviewer_pqaw · 2023-07-27

**Soundness:** 4

**Excitement:**

4: Strong: This paper deepens the understanding of some phenomenon or lowers the barriers to an existing research direction.

**Paper Topic And Main Contributions:**

This paper seeks to improve the calibration of language models, specifically focusing on abstractive text summarization tasks. It explores several calibration methods, including Monte Carlo Dropout, Deep Ensembles, Batch Ensemble, and Spectral-normalized Neural Gaussian Process (SNGP), analyzing their ROUGE performance on three public datasets and sequence-level ECE of their calibrated uncertainty. The paper also delves into the failure patterns of probabilistic methods in NLP community to unveil the significance of selecting appropriate methods based on task setting.


**Questions For The Authors:**

1. What are the definitions of all the terminologies?
2. How are the sequence-level and token-level ECE calculated? Please justify with concrete examples.
3. What is the meaning of the quality in Section 4.3?


**Reasons To Accept:**

- In-depth Analysis: The paper provides a comprehensive comparison of various calibration techniques, including Monte Carlo Dropout, Deep Ensembles, Batch Ensemble, and Spectral-normalized Neural Gaussian Process (SNGP), and adapts these probabilistic methods to LLM setup for summarization task. This detailed analysis includes both generation quality assessment for calibrated model and quality of its uncertainty (length-normalized log-probabilities), which would be invaluable for future research in the field of NLP.
- Clear Presentation: The overall logic flow of the paper is fluent and easy to follow. Both figures and tables are nicely drawn with subtle explanations.

**Reasons To Reject:**

Overall, I like this paper. My main concern is that some details seems missing and need further explanations:
- For the main method (section 3), formal definitions (preliminaries) are necessary before showing equations. For example in Line 186-200, what are the meanings of x, y, t, T? Is u(y|x) a vector or a number (I expect it to be a vector because it is a distribution among the vocabulary space)? Otherwise, readers may find it hard to follow the theoretical part.
- For the evaluation (section 4.2), how is the sequence-level ECE calculated? For example, in line 256-257, if the predicted sequence is “i made fun of a guy walking in giant slippers on father’s day” and the ground truth is “thought a guy’s shoes were the reason he walked funny, turns out he has cerebral palsy.”, does y^ equal y and what is p^? Lack of concrete examples makes it difficult to understand how the accuracy of a prediction (sequence) and the confidence are calculated.
- For the abstention (section 4.3), what is the meaning of the quality in the y-axis of Figure 1? If it is ROUGE, then make it 'ROUGE score vs Abstention', instead of introducing an undefined new term.


**Reproducibility:**

3: Could reproduce the results with some difficulty. The settings of parameters are underspecified or subjectively determined; the training/evaluation data are not widely available.

**Reviewer Confidence:**

4: Quite sure. I tried to check the important points carefully. It's unlikely, though conceivable, that I missed something that should affect my ratings.

---

> ### Author Rebuttal · Authors · 2023-08-29
>
> We are glad that the reviewer liked our paper. We hope that the explanations we provided below will help to answer the questions that you have.
>
> **Definitions of the terminologies at the end of Section 3**
>
> In the formula $u(y|x) := \frac{1}{T} \sum_{t=1}^T \bar{p}(y_t|y_{<t},x)$, $x$ is the input sequence, $y$ is the output sequence, $y_t$ is the $t$-th token of $y$, $T$ is the length of the sequence $y$, i.e. the number of tokens in $y$. We will clarify those terminologies in the camera-ready version.
>
> **How sequence-level ECE is calculated**
>
> The sequence-level ECE is computed by casting the problem into a binary classification task. If the prediction is “i made fun of a guy walking in giant slippers on father’s day” and the ground truth is “thought a guy’s shoes were the reason he walked funny, turns out he has cerebral palsy.”, we say that the prediction is wrong. The probability $\hat{p}$ of a predicted sequence is the production of probabilities of all tokens of that sequence.
>
> **How token-level ECE is calculated**
>
> Tokens make up the space of predictions in token-level ECE. For each input data, we pad or truncate the target sequence such that it has the same length as the predicted sequence. We then say that a predicted token is correct if it matches the target token at the same position in the target sequence. Wang, Shuo, et al. 2020 uses translation error rate to relax the condition that the tokens under consideration need to be at the same position. We will clarify this distinction in the final version.
>
> Wang, S., Tu, Z., Shi, S., & Liu, Y. (2020, July). On the Inference Calibration of Neural Machine Translation. In Proceedings of the 58th Annual Meeting of the Association for Computational Linguistics (pp. 3070-3079).
>
> **Clarifying the meaning of the quality in the y-axis of Figure 1**
>
> This is a great suggestion! Yes, the quality in the figure is the ROUGE-1 score. We agree that using 'ROUGE score vs Abstention' is clearer. We will make the change accordingly and move the ROUGE-1 title to the x-axis of that figure. We also included ROUGE-2 and ROUGE-L results in the appendix.

---

### Official Review · Reviewer_Mqji · 2023-08-05

**Soundness:** 4

**Excitement:**

4: Strong: This paper deepens the understanding of some phenomenon or lowers the barriers to an existing research direction.

**Paper Topic And Main Contributions:**

The paper addresses the challenge of miscalibrated predictive uncertainty in summarization models, which can compromise their reliability in real-world applications. To rectify this issue, the authors perform a study of these methods, analyzing their impact on both uncertainty and prediction aspects of model performance. Additionally, the authors introduce a evaluation protocol tailored to measure uncertainty calibration performance, utilizing domain-specific quality scores like ROGUE. The contributions of the paper include - the adaptation of probabilistic methods to the LLM setup, the evaluation protocol, and the analysis in the experiments. The paper also delves into the identification of failure patterns in well-established probabilistic methods such as Deep Ensembles and Monte Carlo Dropout.


**Questions For The Authors:**

None

**Reasons To Accept:**

- Well written paper, and interesting and informative to read
- Investigation of state-of-the-art probabilistic methods in improving uncertainty calibration in neural summarization models.
-  Clear demonstration of the positive impact of probabilistic methods on summary quality and calibration performance.
- In-depth analysis of failure patterns in widely-adopted probabilistic methods, providing insights for choosing appropriate methods based on data settings.


**Reasons To Reject:**

- The significance and insights from experiment 4.1 are not fully clear, requiring more detailed explanations.
- Although the paper provides good experiments and analysis, it is not entirely novel given the advancements in probabilistic methods for classification tasks.


**Reproducibility:**

4: Could mostly reproduce the results, but there may be some variation because of sample variance or minor variations in their interpretation of the protocol or method.

**Reviewer Confidence:**

3: Pretty sure, but there's a chance I missed something. Although I have a good feel for this area in general, I did not carefully check the paper's details, e.g., the math, experimental design, or novelty.

**Typos Grammar Style And Presentation Improvements:**

line 202
218-220
summarization spelling in 222
standard spelling in figure 2 caption

---

> ### Author Rebuttal · Authors · 2023-08-29
>
> We are happy that the reviewer finds our paper interesting and informative. We thank the reviewer for pointing out the typos in the submission. Corrections to the typos will be made in the camera-ready version. We hope the answers we provide here may address your remaining worries.
>
> **The significance and insights from experiment 4.1 are not fully clear**
>
> The main takeaway from Table 1 is that using a single model probabilistic approach might not necessarily lead to the improvement of the ROUGE scores across every dataset, even if the ECE went down. However in those cases where we observe an improvement it is typically the SNGP method. When it comes to using multiple model probabilistic approaches we observe consistent improvement of ROUGE scores using Deep Ensembles. Deep ensembles are known to be very effective in model calibration and we have demonstrated in our paper that in LLM setting this still holds true. The main conclusion are:
> - Under no time and memory constraints choose Deep Ensembles since they are effective in terms of reducing the ECE and improving the quality of summarization.
> - Calibration methods that are known to be effective in reducing ECE in classification models are not necessarily as effective when adapted to LLMs.
>
> We promise to make these takeaways more clear in the camera-ready version and we thank the reviewer for asking this question.
>
> **Novelty given the advancements in probabilistic methods for classification tasks.**
>
> We agree with the reviewer's observation that our contribution does not lie in introducing novel probabilistic methods. Nevertheless, it's crucial to emphasize that the primary novelty of this paper is to adapt the probabilistic calibration methods from classification to the LLM setting with autoregressive generation and conduct a large benchmarking of the variety of probabilistic calibration methods across different datasets.

---

### Official Review · Reviewer_o7C1 · 2023-08-11

**Typos Grammar Style And Presentation Improvements:** 1. L090
**Soundness:** 3

**Excitement:**

3: Ambivalent: It has merits (e.g., it reports state-of-the-art results, the idea is nice), but there are key weaknesses (e.g., it describes incremental work), and it can significantly benefit from another round of revision. However, I won't object to accepting it if my co-reviewers champion it.

**Paper Topic And Main Contributions:**

The authors proposed a benchmark study about the effectiveness of probabilistic methods in improving the uncertainty quality of the text summarization models, i.e., addressing the problem of miscalibration.
They tested different methods over a T5-base model on three summarization datasets, indicating that Spectral-normalized Neural Gaussian Process combined with Deep Ensembles obtains the best results.

**Questions For The Authors:**

1. L244: What is "beam decoding"? Is it beam search or beam sampling? You should explain it clearly.
2. L542: "Sometimes the summary contains information not present in the article." Who proved it?
3. What is the criterion with which you have selected the following values? 40, 15 and 30 (caption of Table 3).
4. Why did you choose only the T5-base model for the experimental phase?

**Reasons To Accept:**

1. The paper addresses an important topic regarding how to make the output of neural language models more reliable and trustworthy.
2. The problem of miscalibration is understudied in text summarization, meaning that this work could open new research directions toward this topic.

**Reasons To Reject:**

1. The experimental setup is poor. The authors compared several probabilistic methods over a single language model (T5-base). They should evaluate multiple language models of different sizes (e.g., BART-base, BART-large, FLAN-T5-XXL). Furthermore, they should include a long document summarization dataset (and thus a long sequence model such as LED or PEGASUS-X) to make the evaluation analysis more robust.
2. There is not a novel method to address the problem of miscalibration.
3. The authors state to experiment with large language models (LLMs), but in practice, they use T5-base, which is far from being an LLM (like GPT-3).
4. The dataset statistics (i.e., number of source/target words and sentences) should be included to show and demonstrate the difference between the benchmarked datasets.
5. The paper contains many typos (see "Typos Grammar Style And Presentation Improvements" for a non-exhaustive list), meaning the document's drafting has not been curated enough.
6. The authors did not mention how costly each benchmarked method is regarding time and space.

**Reproducibility:**

3: Could reproduce the results with some difficulty. The settings of parameters are underspecified or subjectively determined; the training/evaluation data are not widely available.

**Reviewer Confidence:**

4: Quite sure. I tried to check the important points carefully. It's unlikely, though conceivable, that I missed something that should affect my ratings.

---

> ### Author Rebuttal · Authors · 2023-08-29
>
> Thank you so much for the thoughtful comments and suggestions. We will fix the typos in the camera-ready version. We hope that our response below will address your concerns.
>
> **Evaluate multiple language models of different sizes**
>
> While we appreciate the reviewer feedback, we believe evaluation of the model on such an extensive set of checkpoints goes beyond the scope of the short paper expectation. Additionally we believe that evaluation across multiple architectures should be motivated by finding new information about how the calibration methods work in LLMs. While it would be interesting to see if the model size or instruction tuning can affect the calibration method’s performance, we are not addressing these questions in this work, but rather focus on a general assessment. Finally, we would like to point out that T5 is one of the most widely used transformer based model architectures, so we believe our finding would be generally interesting to the community.
>
>
> **Include a long document summarization dataset**
>
>
> We thank the reviewer for the suggestions and the idea to evaluate the uncertainty calibration in the long document summarization setting. We believe it would be a great research study, and we will definitely consider looking into it in a follow up work. In this paper our aim was to study the application of uncertainty calibration probabilistic deep learning methods to LLMs in a general setting, therefore we have chosen datasets that are in line with the standard benchmark datasets that are usually used in summarization work.
>
>
> **Novel methods to address the problem of miscalibration**
>
>
> We appreciate the reviewer pointing this out. We presume that there might have been a level of confusion, that we would like to correct by explaining the main contributions. We wish to clarify that we don’t claim a novel method that addresses the problem of miscalibration, we do however claim that we have adapted the standard probabilistic calibration methods (that are generally used in the classification tasks) to the LLMs for a sequence generation task and benchmarked them for the first time using newly proposed set of evaluation metrics. We will make sure this is completely clear in the camera ready version.
>
> **T5-base is far from being an LLM**
>
> To the best of our knowledge there is no formal widely accepted definition of the LLM term. LLMs and PLMs have been used interchangeably, T5 is a group of pretrained LMs, there are currently more modern LLMs that have sizes similar to T5 models. In several published and academically recognised works T5-base checkpoint is regarded as an LLM and we share the same understanding.
> It would definitely be interesting to perform the study using LLMs like GPT-3. More infrastructure work will be required to employ uncertainty methods considered in the paper to those LLMs. We leave it open for future study.
>
>
> **The dataset statistics**
>
> We thank the reviewer for the idea to include this information into the paper. For now we provide the following statistics on the word account and we will later add it into the paper (together with the information about the number of sentences):
>  - The CNN/DailyMail dataset has on average 687 words in the input document and around 53 words on average in the summary.
>  - XSum has 375 words on average in the input document and 21 words in the summary.
>  - Reddit Tifu has 433 words on average per input document and 23 words per summary.
>
> **Time and space complexity of each method**
>
> Thank you for the question! We will clarify the complexity of each method compared to the time complexity $T$ and space complexity $M$ of the base model as below.
>
>  - SNGP, BE: time complexity $O(T)$, space complexity $O(M)$
>  - MCD, SNGP+MCD: time complexity $O({num \textunderscore mc \textunderscore  samples}  \times  T)$, space complexity $O(M)$.
>  - DE, SNGP+DE: time complexity $O(ensemble \textunderscore size \times T)$, space complexity $O(ensemble \textunderscore size \times M)$.
>
>
> **What is "beam decoding"?**
>
> We apologize that the term was not clear and we will fix this in the camera ready version, and refer to the term as "beam search decoding", however we would like to point out that it is not uncommon to refer to "beam search decoding" algorithm as "beam decoding", for example see "Beam decoding with controlled patience" by
> J Kasai, K Sakaguchi, RL Bras, D Radev, Y Choi, NA Smith.
>
>
> **In XSUM dataset, the summary might contain information not present in the article**
>
> We thank the reviewer for pointing this out, we agree that the sentence can be phrased differently and we can provide an example from XSum that confirms the claim, which we promise to include this into the final version of the paper:
>
> - Input document:
> "Sustainable Shetland, a group opposed to the development, has announced it intends to seek a judicial review of the Scottish government's decision to approve the development. The wind farm would be the third biggest in Scotland, run by community company Viking Energy. Energy Minister Fergus Ewing granted consent for the scheme in April. Protesters claim the development is too big and would blight the landscape. Supporters argue it would raise money for the islands, create jobs and help meet renewable energy targets. The 370MW wind farm is aimed at powering more than 175,000 homes despite Shetland having a population of about 22,000. It is estimated the wind farm could bring about  £30m annual income for the local community."
>
> - Summary:
> "Controversial plans to build a **103-turbine** wind farm in the centre of Shetland could face a legal challenge."
>
> **"103-turbine"** only appears in the summary and not in the text.
>
> Please also refer to the paper named  "On Faithfulness and Factuality in Abstractive Summarization" by
> Joshua Maynez, Shashi Narayan, Bernd Bohnet, Ryan McDonald, where authors discuss this phenomenon.
>
> **Criterion for selecting the values 40, 15 and 30 in the Table 3**
>
> The values were selected as round numbers on the SoTA performance circa 2020 when NLG fluency was human-like: https://paperswithcode.com/sota/text-summarization-on-x-sum.
>
> **Why only the T5-base model is used for the experimental phase**
>
> Our codebase is based on the open-sourced T5X framework. Since we must load different sets of the model’ parameters into acceleration devices for uncertainty approaches like Deep Ensemble, the memory requirement is substantially higher than for the base model. As a result, we decided to use the T5-base, which works smoothly across methods and tasks.

---

### Official Review · Reviewer_X5Gn · 2023-08-12

**Soundness:** 4

**Excitement:**

4: Strong: This paper deepens the understanding of some phenomenon or lowers the barriers to an existing research direction.

**Paper Topic And Main Contributions:**

The paper addresses the research gap that uncertainty calibration has not been well studied in summarization, especially with probabilistic neural summarization methods.  The paper investigates the effectiveness of the classic approaches and the state-of-the-art methods of probabilistic methods in improving the uncertainty quality of the neural summarization models. The paper also proposes the evaluation protocol including: ROUGE for quality improvement, sequence-level Expected Calibration Error (ECE) for calibration improvement, rank correlation for quality and predictive confidence as they changed with calibration, and quality vs abstention curve that measures change in quality as certain fraction of uncertain prediction examples are removed.

**Questions For The Authors:**

A. Why isn't the AESLC dataset used for this study?

B.  The "quality vs abstention curve" is essentially the same as the "performance vs data retention curve" as proposed by Filos et al. 2019, isn't it?  If so, it's better to cite and include the reference:

Angelos Filos, Sebastian Farquhar, Aidan N. Gomez, Tim G.J. Rudner, Zachary Kenton, Lewis Smith, Milad Alizadeh, Arnoud De Kroon, and Yarin Gal. A systematic comparison of Bayesian deep learning robustness in diabetic retinopathy tasks, 2019. ISSN 23318422.

**Reasons To Accept:**

The paper performs systematic evaluation of the probabilistic methods in uncertainty calibration in neural summarization models. The evaluation is performed on classic and state-of-the-art methods, on three datasets, and with evaluation metrics of ROUGE, ECE, correlation between quality and predictive confidence, and quality vs abstention rate.

**Reasons To Reject:**

1. It is not quite reassuring to see that the authors use the "average" performances from the three datasets (as in Tables 1 and 2) to prove the effectiveness of the probabilistic models. The average rank as in Table 1 is more reasonable. And yet, the average rank is not used in Table 2, which needs explanation.

2. The evaluation is systematic, covering quality (with ROUGE), calibration (sequence-level and token-level ECE), quality vs predictive confidence, and quality vs abstention rate.  However it is a bit farfetched to say it's novel.

**Reproducibility:**

4: Could mostly reproduce the results, but there may be some variation because of sample variance or minor variations in their interpretation of the protocol or method.

**Reviewer Confidence:**

3: Pretty sure, but there's a chance I missed something. Although I have a good feel for this area in general, I did not carefully check the paper's details, e.g., the math, experimental design, or novelty.

**Typos Grammar Style And Presentation Improvements:**

Shouldn't the "access" as in lines 208 and 247 be "assess"?

Above line 272, caption of Figure 2: "standart" should be "standard".

Line 081: "latecy" ?

Line 202: "we" should be "We".

Notes need to be added to Table 1 and 2 about the meaning of the "bolded" and the "underlined" format.

---

> ### Author Rebuttal · Authors · 2023-08-29
>
> We thank the reviewer for helpful comments and pointing out the typos. We will incorporate the suggestions into the camera ready version. We hope that our responses below will address your concerns.
>
>
> **Average rank for table 2**
>
> We agree with the reviewer that having an average rank in Table 2 would make the results more consistent and clear, therefore we will add the average rank metric to Table 2 for the camera ready version.
>
>
> **Evaluation novelty**
>
> Although we acknowledge the reviewer's observation that the individual calibration approaches are not inherently groundbreaking, we believe the novelty of our paper lies in implementing and adapting these calibration techniques inside the LLM framework and proposing the evaluation metrics for these methods within the context of the LLM domain.
>
>
> **Studying the AESLC dataset**
>
> We are originally interested in standard tasks without too brief summarization. Summaries in the AESLC dataset are the subjects of emails, which are a bit short and vague. We agree that studying uncertainty methods on this dataset is a worthwhile task. We want to leave it open for further investigation.
>
> **Relationship between “quality vs abstention curve” and “performance vs data retention curve”**
>
> We appreciate the reviewer bringing up the paper by Filos et al. 2019. In Filos et al. 2019 data is retained based on predictive entropy. In our paper, we employ log probability as the metric for data retention. In our camera-ready version, we will properly cite the Filos et al. 2019 paper and explain this distinction.

---

### Meta-Review · Area_Chair_QfMN · 2023-09-17

**Recommendation:** 3

**Metareview:**

The authors studied the problem of uncertainty calibration on the summaries from PLM. This is an interesting problem to NLP community.
The authors investigate different probabilistic methods in addressing the problem. The experiments were on T5-base and results showed that these methods can improve the uncertainty quality.
A weakness of this paper is that evaluation was carried out on T5-base. It is not an LLM in light of the current progress. Nevertheless, the results on T5-base look thorough.

---

### Decision · Program_Chairs · 2023-10-07

**Decision:**

Accept-Findings

**Comment:**

The authors studied the problem of uncertainty calibration on the summaries from PLM. This is an interesting problem to NLP community.
The authors investigate different probabilistic methods in addressing the problem. The experiments were on T5-base and results showed that these methods can improve the uncertainty quality.
A weakness of this paper is that evaluation was carried out on T5-base. It is not an LLM in light of the current progress. Nevertheless, the results on T5-base look thorough.